# Impella 5.0/5.5 Implantation via Innominate Artery: Further Expanding the Opportunities for Temporary Mechanical Circulatory Support

**DOI:** 10.3390/jcm11195917

**Published:** 2022-10-07

**Authors:** Stephanie Bertolin, Giulia Maj, Corrado Cavozza, Astrid Cardinale, Alberto Pullara, Andrea Audo, Federico Pappalardo

**Affiliations:** 1Department of Cardiothoracic and Vascular Anesthesia and Intensive Care, AO SS Antonio e Biagio e Cesare Arrigo, Via Venezia, 16, 15121 Alessandria, Italy; 2Department of Cardiac Surgery, AO SS Antonio e Biagio e Cesare Arrigo, Via Venezia, 16, 15121 Alessandria, Italy; 3Department of Cardiology, AO SS Antonio e Biagio e Cesare Arrigo, Via Venezia, 16, 15121 Alessandria, Italy

**Keywords:** Impella 5.0/5.5, innominate artery, mechanical circulatory support

## Abstract

When axillary/subclavian arteries are not suitable because of size or anatomy, alternative access for the Impella pump 5.0/5.5 via the innominate artery allows circulatory support and eventually de-escalation from VA-ECMO to isolated left-side support. Moreover, less invasive surgery without the need to open the pericardium reduces the risk of RV dysfunction and bleeding. Finally, upper body strategies allow early rehabilitation during support, which is associated with improved survival in cardiogenic shock.

## 1. Introduction

The microaxial blood pump Impella 5.0/5.5 is effective to provide temporary circulatory support, allowing heart recovery and improving outcomes of cardiogenic shock [1]. Moreover, transitioning patients with acute cardiogenic shock from VA-ECMO to Impella 5.0/5.5 minimizes ECMO-related complications, including bleeding and RV dysfunction. This two-step strategy is effective, especially in INTERMACS 1 patients, as a bridge to durable LVAD. Indeed, after the resuscitation phase with a short VA-ECMO support period, recovery of end-organ function, revaluation of right ventricular performance, and optimization of any modifiable factor during Impella 5.0/5.5 support, durable LVAD can be implanted with reasonable chances of success [2].

Upper body (axillary/subclavian) implantation offers multiple advantages compared to the femoral approach, including feasibility in peripheral vascular disease, mobilization, and early rehabilitation on support notwithstanding the stability of the catheter [3,4]. Indeed, the last Impella 5.5 was designed for axillary implantation only, with enhanced deliverability and torque response. When the axillary arteries are not suitable (small size <7 mm, severe atherosclerosis, patent LIMA-LAD, anatomical challenges), direct implant in the ascending aorta is a standardized technique. We describe a further implantation option via the innominate artery with upper ministernotomy, especially in the context of de-escalation from VA-ECMO, and describe its potential benefits (Figure 1).

A 71-year-old obese man with ischemic cardiomyopathy and chronic refractory angina treated with the Reducer device was admitted to a spoke hospital with acute antero-lateral STEMI. Sudden cardiac arrest required percutaneous femoro-femoral VA-ECMO implantation as eCPR. However, despite limb reperfusion, the patient had peripheral ischemia of the cannulated limb in the context of severe PAD: a sheathless IABP was implanted in the contralateral limb. On day 2, de-escalation from VA-ECMO to Impella 5.0 via the axillary artery was planned, but access was not suitable at surgery because of severe calcific atherosclerosis. The innominate artery approach was performed. VA ECMO was subsequently weaned 24 h later, and the patient was successfully weaned from Impella on day 7 at bedside.

## 2. Technique

Under general anesthesia and systemic heparinization, an upper partial sternotomy with extension to the right third intercostal space (J-shaped incision) is performed. After visualization of thymic residuals without pericardial opening, the innominate vein is identified and retracted inferiorly with umbilical tape, avoiding injury to the right recurrent laryngeal nerve. Then, the proximal part of the innominate artery is exposed and dissected up to its bifurcation. A 10 mm dacron graft (Gelweave Vascutek) is sutured using 5-0 prolene end-to-side to its anterior surface after partial clamping and is tunneled to the supraclavicular space (Figure 2 and Figure 3). Thereafter, a standard Impella 5.0/5.5 procedure is performed. The sheath is inserted into the graft and secured with provided graft locks to prevent blood loss during implantation. Then, after removal of the vascular clamp on the graft, a 0.035 Inch guidewire is inserted into the sheath to the left ventricle, crossing the aortic valve under fluoroscopic and transesophageal echocardiography guidance. Finally, the device is progressively pushed into the graft, introduced into the innominate artery to the left ventricle cavity, and, after guidewire removal, turned on. The graft lock is then released, and the introducer removed, slid out of the graft, and peeled it away. The sheath is blocked distally and fixed at the skin to secure the driving cable and pump in position. The vascular graft is trimmed at the skin level and is pushed under the surgical wound in order to avoid any external exposure. A mediastinal chest tube is placed, and the chest is closed (Figure 4). To remove the pump, under general or local anesthesia, the Impella catheter is withdrawn without surgical reopening of the upper partial sternotomy. Only the superficial sutures are cut to expose the graft. The sutures around the prosthesis that secure the catheter to the graft are cut, and the pump is removed. Finally, the vascular graft is ligated, oversewn, and buried in the subcutaneous tissue.

We envision multiple advantages of this strategy compared to median sternotomy and ascending aorta anastomosis in the setting of VA ECMO, both for unloading and de-escalation:-Reduction of the surgical trauma is pivotal in patients on ECMO to reduce bleeding, assuming that anticoagulation and ECMO-associated coagulopathy are systematically considered. Many patients might also receive DAPT because of the ischemic etiology of cardiogenic shock and recent PCI.-Avoidance of opening the pericardium might warrant better preservation of right ventricular function, allowing safe de-escalation to isolated left ventricular support from the ECPella configuration.-Upper body strategies are pursued due to their benefits for patient management: feasibility in severe peripheral artery disease, easy mobilization on support, and early rehabilitation. Indeed, as soon as the hemodynamic status improves, the patient is able to get out of bed, rest in an armchair, walk around the room, undergo phydiotherapy, and start oral feeding.-Bedside weaning without reopening of the surgical route is an option.-Proximity to the aortic valve overcomes the technical issues encountered via the axillary approach (calcification, tortuosity under the clavicle, small vessel size).

Data on RHF occurrence when transitioning from VA-ECMO to Impella 5.0/5.5 are lacking, and RV dysfunction in this setting should be further investigated. However, data on the INTERMACS 1 profile patients in the context of durable LVAD might be transferred to the cardiogenic shock setting with temporary MCS. In fact, RHF represents a major complication and the most important cause of early death after LVAD implantation in critically ill patients despite normal echocardiographic RV parameters. Furthermore, evaluation of the RV during VA ECMO is challenging, and de-escalation strategies should consider RVF as a consistent risk. Recent studies suggest that Less Invasive Surgery (LIS) via small anterior thoracotomy and upper hemisternotomy might beneficially impact RHF occurrence compared to standard full median sternotomy, preserving RV function. The difference between the two operative approaches becomes more pronounced as the perioperative RVF risk increases. The background for the LIS benefits includes not only decreased CPB time and less manipulation of the heart, but also a restraining effect of intact pericardium on right chambers; this is even more relevant when the right heart is loaded by the potent left-side support. Indeed, opening the pericardium affects RV contractility and RV geometry and allows for RV dilatation in response to increased preload. Moreover, keeping the pericardium closed reduces abnormal septal motion, also preserving LV geometry and the risk of suction [5].

Finally, learning from the TAVI space, in conditions that restrict subclavian access, transcarotid insertion offers many advantages compared to apical or transaortic strategies: in fact, not including instrumentation of the aortic arch, cerebral embolization events can be reduced. To date, many studies confirm the safety of this alternative approach regarding early and long-term cerebrovascular complications [6].

This technique is also valuable in pediatric patients, as small peripheral vessels cannot accommodate the device [7].

## 3. Conclusions

Alternative access for the Impella 5.0/5.5 pump via the innominate artery adds a further option in the armamentarium of implantation techniques, as it maintains all the beneficial features of axillary cannulation while avoiding full sternotomy and pericardiectomy. As de-escalation and ambulation strategies are leading the pathway to improved outcomes of cardiogenic shock, we envision a space for this technique in the surgical portfolio to manage hostile axillary arteries, especially in patients with VA-ECMO.

## Figures and Tables

**Figure 1 jcm-11-05917-f001:**
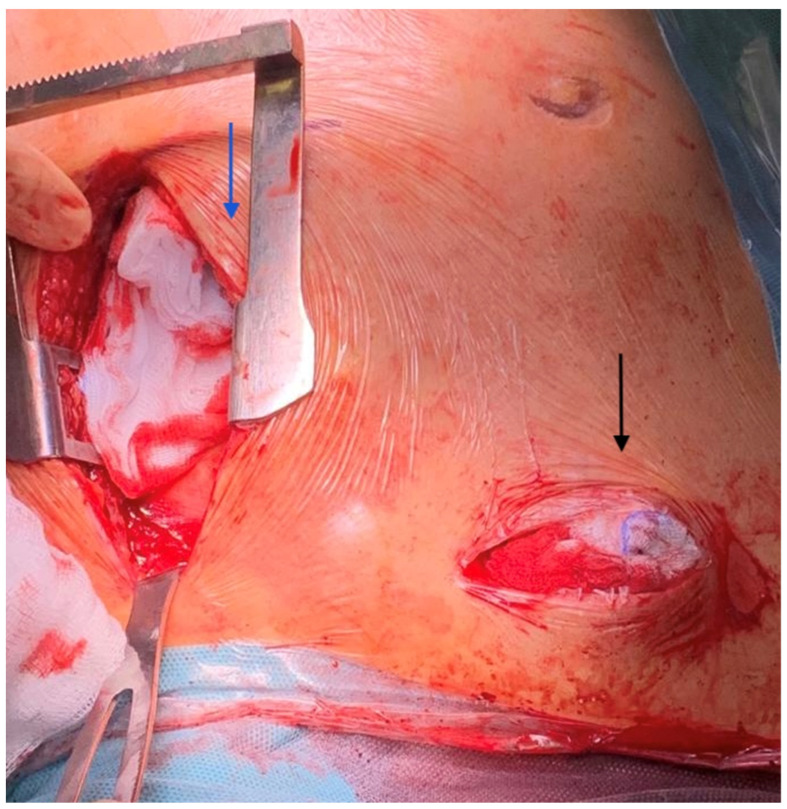
Attempt to approach the right axillary artery failed for severe atherosclerosis (black arrow). Conversion to trans-innominate implantation via ministernotomy (blue arrow). J-shaped incision is performed.

**Figure 2 jcm-11-05917-f002:**
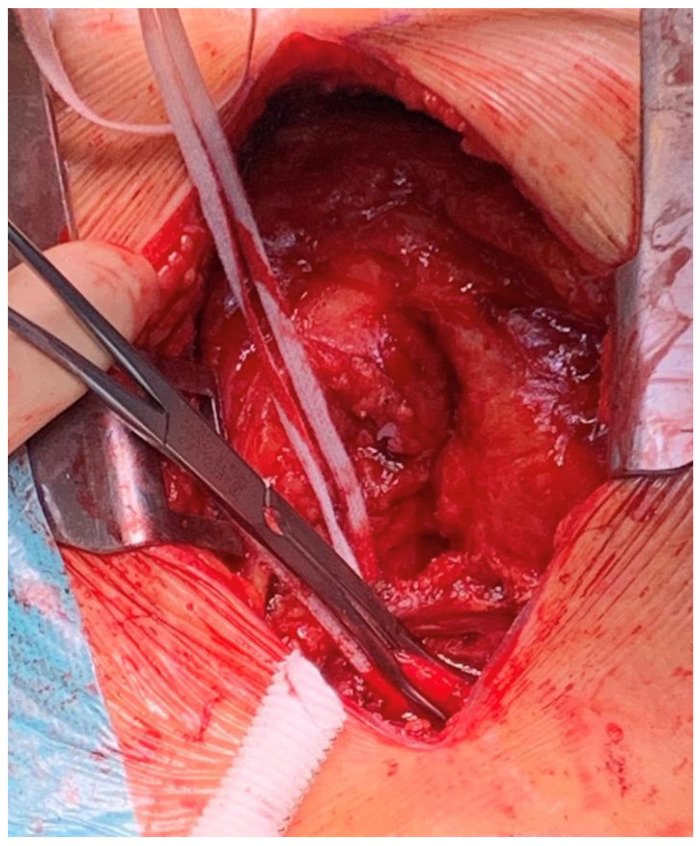
Exposure of the innominate artery and partial clamping for vascular graft anastomosis.

**Figure 3 jcm-11-05917-f003:**
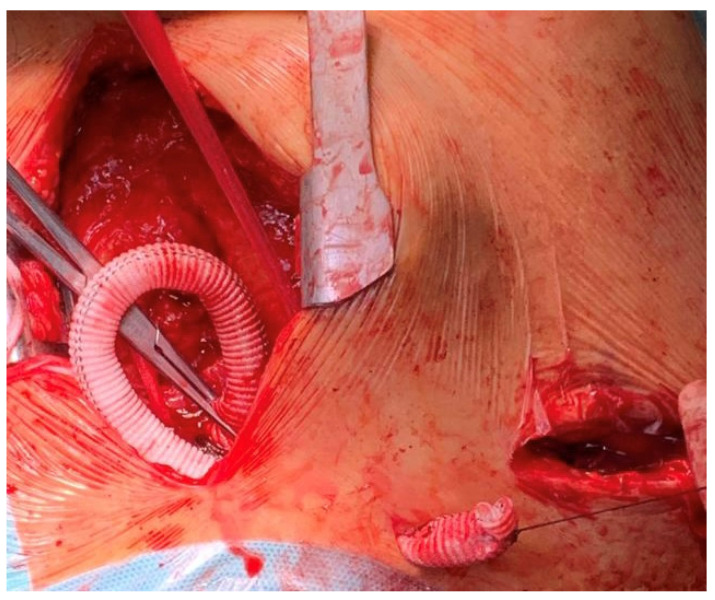
Tunneling of the graft via the subcutaneous supraclavicular space.

**Figure 4 jcm-11-05917-f004:**
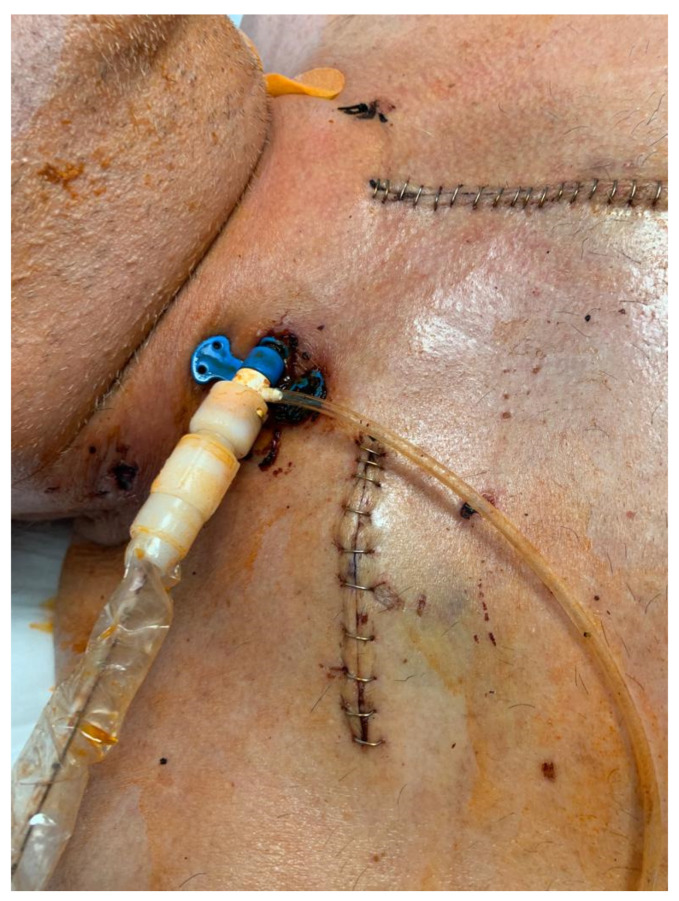
Final result with repositioning sheath secured to the skin. The graft is trimmed to sit below skin level.

## Data Availability

Not applicable.

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
