# Peer review of "Impella 5.0/5.5 Implantation via Innominate Artery: Further Expanding the Opportunities for Temporary Mechanical Circulatory Support"

_jcm, 2022, doi:10.3390/jcm11195917_

Round 1

Reviewer 1 Report

Dear authors,

thank you for giving me the opportunity to review your manuscript. It is very well written and illustrated. My only comments are that based on your discussion you should discuss a manuscript that has been published on the concept of bridging to LVAD and occurrence of RV failure (Bertoldi et al. J Crit Care 2020). 

Again, thank you for the opportunity to review your case report

Author Response

Thanks for your appreciation and your interesting suggestions. We have included in the Introduction the concept of transitioning from VA-ECMO to Impella 5.0 as bridge to LVAD and the role of Impella 5.0 support period in improving any modifiable risk factor before LVAD implantation, including right ventricular performance. 

Reviewer 2 Report

In this case report article, Bertolin et al nicely describe the Impella Implantation technique through the innominate artery. The article is well written and provides an accurate description of this specific technique, so the authors should be congratulated. I have some considerations:

1. Major issues:

1.1. Considering that it is a case report article, the readers would appreciate at least a summary or small presentation of the clinical case, to properly put in context the use of this access in this specific case.

2. Minor issues:

2.1. There are several abbreviations used directly in the first appearance in the text (e.g., LIMA, DAPT, PCI, RHF, RV, MCS, RHF, RVF, etc..). While most of them would be familiar to readers because of their widespread use (e.g., LIMA, DAPT, PCI, TAVI), I suggest not using the abbreviate form in their first appearance in the less common ones (e.g., RHF, RVF, MCS).

2.2. “Upper body strategies are still pursued with their well acknowledged benefits on patient management”. Please, I suggest citing some examples of these acknowledged benefits.

Author Response

Dear Reviewer, thanks for the appreciations and for your suggestions.

1)Major issue: We have inserted a small presentation of the clinical case and the rationale of the approach chosen. Since axillary arteries were not suitable for severe atherosclerosis, a direct aortic approach would have required performing a full sternotomy with high risk of bleeding and surgical opening of the pericardium with risk of right ventricular dysfunction, which is one of major causes of death in INTERMACS I patient in MCS.

2) Minor issues:

  • 2.1 We have inserted the section "Abbreviations" to facilitate the readers.
  • 2.2 We have expanded the topic with examples.

Reviewer 3 Report

I read with interest the article entitled “Impella 5.0/5.5 Implantation via innominate Artery: Further Expanding the Opportunities for Temporary Mechanical Circulatory Support” and I think it could be of some interest to both cardiologists and cardiac surgeons. Temporary left ventricle assist devices are increasingly employed in several clinical scenarios and management of vascular access remains a challenging issue. Impella 2.0 and Impella CP can be placed with a standard percutaneous approach and femoral, axillary and carotid access have been all demonstrated to be safe and sound. Because of the larger diameter of the pump’s motor, Impella 5.0 and 5.5 have been designed for a trans-axillary placement after surgical exposition and isolation of the artery. Sometimes axillary artery stenosis (MLD<7mm) or extreme tortuosity does not allow to advance of the catheter through this route safely and direct transaortic placement can be tried. Nevertheless, if Impella 5.0 or 5.5 are needed, a severe cardiogenic shock is probably faced and in a similar setting this kind of invasive approach can be detrimental for an extremely fragile patient. For this reason, an access through the innominate artery obtained with a small chest incision is an attractive alternative.

The paper describes each stage of the procedure; the text is well written and easy to understand. I have only one doubt that hopefully can be solved by the authors with a brief explanation. It is written, “Bedside weaning without reopening of the surgical route is still an option”. Personally, it’s not clear how the Impella catheter can be withdrawn without a surgical revision, taking into account that the graft conduct used for the advancement has to be removed and puncture in the artery has to be managed with some haemostatic technique.

Author Response

Thanks for the suggestion, we have detailed the removal technique to explain the bedside approach.

Reviewer 4 Report

In the present case report authors describe an alternative method for mechanical circulatory support (Impella 5.0/5.5) via innominate artery. Authors state that in case of unsuitable axillary or subclavian artery presented method provides a good option. Authors have a clear description of the procedure. Only criticism is the abstract which is like a list. I think it would benefit from revision to be more fluent. Also figure 1 could be more informative.

Author Response

Dear Reviewer, thanks for sharing some important points of the topic.

We have implemented your suggestions both in abstract and in figure 1.

Reviewer 5 Report

Thank you for the opportunity to review this manuscript.

In this case report, the authors describe the use of alternative access for the Impella pump. The manuscript is well written, thorough, and describes the use of an Impella implanted within the via the innominate artery.  

Comments

1.     In the Introduction, the authors list a patent LIMA as a contraindication for Impella implantation in the left axillary artery. I assume the authors are referring to a patent LIMA that is grafted to the heart, instead of any patent LIMA as this would significantly limit the patient population that would be appropriate for axillary implantation. This should be clarified.

2.     Under the technique section, the authors describe their approach to gaining access to the innominate artery and setup for the Impella, which is followed by “ Thereafter, a standard Impella 5.0/5.5 procedure is performed.” As not all readers may be familiar with the procedure for placement of Impella, I would suggest the authors include a brief description of the procedure or include a reference that describes the standard implantation of the Impella.

3.     It may also be of interest to readers if the authors discuss the removal of the Impella and if there are any specific considerations for this approach.

4.     The authors discussed the benefits of this approach in detail, but do not discuss the limitations or risks associated. This should be included in the Discussion as well.

5.     The figures provided are appropriate and complement the information provided in this study. 

Author Response

1)With patent LIMA-LAD we intended patent left internal mammary arterial graft to left anterior descending coronary artery. In patients with a patent left internal mammary artery coronary bypass graft, Impella 5.0 placement via Innominate artery is an alternative approach in order to prevent myocardial ischemia.

2)We have described the standard procedure for Impella 5.0/5.5 placement.

3)We have described the removal technique, detailing the bedside approach opportunity

4) Thank you for this comment: the risks of this technique are related to the opportunity of accessing the axillary (which would be the best option) and compared to the alternative (ascending aorta) option available. Therefore, we report on the limitation of risks associated with direct ascending aorta cannulation with full sternotomy and pericardial opening, especially during VA ECMO support. This is the key message of the paper.

5) Thanks for your appreciation about the figures.